# Profiles of Active Transportation among Children and Adolescents in the Global Matrix 3.0 Initiative: A 49-Country Comparison

**DOI:** 10.3390/ijerph17165997

**Published:** 2020-08-18

**Authors:** Silvia A. González, Salomé Aubert, Joel D. Barnes, Richard Larouche, Mark S. Tremblay

**Affiliations:** 1Healthy Active Living and Obesity Research Group, Children’s Hospital of Eastern Ontario Research Institute, Ottawa, ON K1H 8L1, Canada; saubert@cheo.on.ca (S.A.); j@barnzilla.ca (J.D.B.); richard.larouche@uleth.ca (R.L.); mtremblay@cheo.on.ca (M.S.T.); 2School of Epidemiology and Public Health, Faculty of Medicine, University of Ottawa, Ottawa, ON K1G 5Z3, Canada; 3Faculty of Health Sciences, University of Lethbridge, 4401 University Drive, Lethbridge, AB T1K 3M4, Canada

**Keywords:** cycling, walking, health promotion, policy, latent profile analysis, surveillance

## Abstract

This article aims to compare the prevalence of active transportation among children and adolescents from 49 countries at different levels of development. The data was extracted from the Report Cards on Physical Activity for Children and Youth from the 49 countries that participated in the Global Matrix 3.0 initiative. Descriptive statistics and a latent profile analysis with active transportation, Human Development Index and Gini index as latent variables were conducted. The global average grade was a “C”, indicating that countries are succeeding with about half of children and youth (47–53%). There is wide variability in the prevalence and in the definition of active transportation globally. Three different profiles of countries were identified based on active transportation grades, Human Development Index (HDI) and income inequalities. The first profile grouped very high HDI countries with low prevalence of active transport and low inequalities. The second profile grouped low and middle HDI countries with high prevalence of active transportation and higher inequalities. And the third profile was characterized by the relatively high prevalence of active transportation and more variability in the socioeconomic variables. Promising policies from countries under each profile were identified. A unified definition of active transportation and contextualized methods for its assessment are needed to advance in surveillance and practice.

## 1. Introduction

The world is experiencing a crisis of physical inactivity with almost 80% of adolescents not achieving the recommended 60 min of daily moderate to vigorous physical activity for health [1]. In this context, transportation, as a daily necessity to move from one place to another, represents a promising domain to promote the accumulation of physical activity in children and adolescents in a convenient and habitual manner [2]. Specifically, active transportation to/from school is an opportunity to integrate physical activity into children’s and adolescent’s routines [3].

Active transportation comprises non-motorized travel modes like walking, cycling or riding a scooter, among others [4]. The use of these active modes leads not only to health benefits such as greater levels of cardiorespiratory fitness [3,5] and better cardiometabolic health indicators [6] among children who actively commute, but also to other co-benefits, such as better mental health outcomes [7,8], greater interaction with their environment [9], and reduced transportation-related emissions and pollution [10]. Despite these benefits, current evidence suggests that this behaviour is declining in many countries [11].

In the same way that physical inactivity prevalence varies widely across countries [1], a wide variation in active transportation could be expected. These variations represent an opportunity to identify those countries that are succeeding with active transportation behaviours, and those that require action to increase active transportation or prevent a decline in this behaviour. However, to the best of our knowledge, the few international comparisons of data on active transportation among children and adolescents include mostly small groups of countries or the availability of national representative data is limited [11,12,13]. Therefore, the Global Matrix 3.0 of Report Card grades on physical activity among children and youth provides an opportunity to describe and examine the global situation of active transportation. For the first time, 49 countries from all continents reported data on an active transportation indicator at the national level [14]. The aims of this study were to compare the prevalence of active transportation among children and adolescents from 49 countries participating in the Global Matrix 3.0, to identify a set of profiles to group the countries according to their prevalence of active transport and sociodemographic variables, and to discuss policies and practices implemented across different countries to increase active transportation.

## 2. Materials and Methods

The Global Matrix 3.0 was an international initiative released in 2018 and led by the Active Healthy Kids Global Alliance (AHKGA). This project brought together 513 researchers and physical activity leaders from 49 countries around the world [15]. All the participating countries followed a harmonized process to develop Report Cards on the physical activity of children and youth. A detailed description of the countries’ involvement and the process to develop the Report Cards has been published elsewhere and is briefly described here [14].

In each country, National Report Card Committees gathered the best and most recent national surveillance data available up to 2018 to inform and grade ten specific indicators related to physical activity among children and adolescents: Overall Physical Activity, Organized Sport and Physical Activity, Active Play, Active Transportation, Sedentary Behaviours, Physical Fitness, Family and Peers, School, Community and Environment, and Government [14]. The analyses presented in this paper are focused on the Active Transportation indicator.

According to the benchmarks proposed by AHKGA to harmonize and guide the development of the Report Cards, the Active Transportation indicator was described as the “percentage of children and youth who use active transportation to get to and from places (e.g., school, park, mall, friend’s house)” [14]. Report card leaders were instructed to inform this indicator by the best, preferably nationally representative, data available for children and adolescents between five and 17 years, and a grade was assigned according to the prevalence following a common rubric established by the AHKGA (Table 1).

The prevalence of active transportation reported by each country and the related details presented in each Report Card, including policies, practices, strategies to improve the grade and research gaps, were extracted from the Report Cards and from related publications in English, Spanish or French, including brief reports, posters and peer-reviewed articles. These publications were reviewed, and relevant information was summarized by two of the authors of this manuscript. Based on the grades provided, numerical equivalents were assigned (Table 1), and average estimates of the grades for active transportation were calculated at the global level and by groups of countries according to their level of development determined by the Human Development Index (HDI). The HDI is a composite index created by the United Nations Development Programme (New York, NY, USA) to rank countries based on key dimensions of human development such as education, life expectancy and gross national income per capita [16]. HDI ranges from 0 to 1 and for the present analysis we used the continuous index and a categorical variable that classified countries in three categories: low and medium (HDI < 0.70), high (HDI ≥ 0.70 to <0.80) and very high (HDI ≥ 0.80) [16]. It was included as a variable of interest in this analysis based on the variability in active transportation observed across HDI clusters in previous analysis of the Global Matrix [14]. Also, the Gini index for each country was retrieved from the World Bank estimates. The Gini index provides a measure of inequality in income distribution. It ranges from 0 (perfect equality) to 100 (perfect inequality) [17]. The Gini index was included in this analysis considering previous international evidence that has shown that income inequality is a relevant variable related to physical activity levels and taking into account the importance of socioeconomic inequalities in transport as an essential activity for economic and social development [18,19].

A latent profile analysis (LPA) was conducted to identify groups or profiles of countries based on the numerical grades for active transportation and the two sociodemographic variables at the country level, the HDI and the Gini index. LPA is a probability-based statistical procedure that allows to identify classes or profiles that group observations sharing similar patterns of the variables of interest [20]. The analysis was performed to look for the best model solution for one to five possible profiles. Models were compared to choose the solution with the best fit based on the Akaike information criterion (AIC), sample-adjusted Bayesian information criterion (SABIC) and the bootstrapped likelihood ratio test (BLRT) as indicators of model fit. All statistical analyses were performed using SAS 9.4 (SAS Institute, Cary, NC, USA) and R (version 3.4.1, The R Foundation for Statistical Computing, Vienna, Austria). The tidyLPA package [21] was used for the LPA.

## 3. Results

A total of 47 countries (96%) in the Global Matrix 3.0 had sufficient evidence (determined by each country’s National Report Card Committee) on active transportation to assign a grade. The grades ranged from “A−” in Japan, Nepal and Zimbabwe to “F” in Chile (Table 2). The global average for active transportation was “C”. The average grade by HDI was “C+” for low to medium HDI countries, “C” for high HDI countries and “C−” for very high HDI countries, as previously reported by Aubert et al. [14]. The HDI of the included countries varied from 0.448 in Ethiopia to 0.985 in Jersey. According to the Gini index, the country with the most unequal distribution of income was South Africa with a Gini index of 63, while Slovenia had the lowest inequality score, with a Gini of 25.4 (Table 2).

Table 3 presents the prevalence and rationales behind the grades for each country, as well as the sources and characteristics of the information reported. Active transportation among children and adolescents varied between 15% in Chile and 86% in Japan and Nepal. Among the countries that assigned a grade for active transportation, 83% (*n* = 39) did not provide details of the prevalence stratified by sex. In the majority (62%) of countries that reported data by sex, the prevalence of active transportation was slightly higher for males. More than half of the countries (65%) reported data for both children and adolescents, however, the age groups included varied from one country to another. Most countries (87%) only included data on school trips, and only two countries (Ecuador and the United States) clearly reported active transportation to other destinations. Regarding the direction of the trips, about half of the countries (49%) reported active transportation to and from school or other destinations. In more than half of the countries (65%), the frequency of active transportation reported was not clear. The most common frequencies reported were “daily” (*n* = 3), “typically” or “usually” (*n* = 3) and “on a regular basis” (*n* = 2). Regarding the source of information, 64% (*n* = 30) of the countries used data from surveys and studies with national representativeness, 8.5% (*n* = 4) used local studies, and 19% (*n* = 9) used both local and national studies. International surveys such as the Global School-Based Student Health Survey (GSHS) [22] and the Health Behaviour of School-aged Children (HBSC) [23] were among the sources of information in seven countries.

The best LPA model grouped the Global Matrix 3.0 countries into three profiles according to the grades for active transportation, the HDI and the Gini index. The three-profile model had the best fit statistics according to the criteria proposed by Nylund et al. for model selection [24]. The preferred model showed the lowest values for the AIC (359.8), SABIC (331.1) and the BLRT (24.8), and a significant *p* value for the BLRT (*p =* 0.041). Table 4 shows the descriptive statistics for the latent variables among the three profiles identified. In profile 1 (*n* = 25) 72% of the countries had active transportation grades below “C”, 96% of the countries had a very high HDI, and 72% had relatively low Gini indices (below 40). In profile 2 (*n* = 7), 85% of the countries had active transportation grades equal to or greater than “C”, all of them had a low to medium HDI and 43% had Gini indices above 40. In profile 3 (*n* = 17), 94% of the countries had active transportation grades equal to or greater than “C”, 53% had a high HDI and 35% had a very high HDI, and 47% had Gini indices above 40. For countries with missing values in any of the variables of interest, the LPA assigned a profile based on the values available for the remaining variables. Figure 1 presents a plot of the scaled data for the three profiles.

The availability of details related to active transportation in the report cards, beyond the reported prevalence, varied across countries. Table 5 summarizes the information provided by countries in terms of practices and policies, strategies proposed to improve the grades and research gaps identified by expert groups in each country. Twenty-four countries provided at least one of these details. The policies and practices identified by the expert groups included school siting policies, transport policies that prioritize active modes of commuting, walking challenges and special events, and multi-component programs that comprise educational strategies, enforcement of regulation to improve traffic safety, and providing infrastructure and resources at several levels (children, teachers, schools and communities). The most common topics in the strategies proposed to improve the grades were improving safety conditions, providing supportive infrastructure, developing informational and education strategies, and involving parents, schools and communities in the promotion of active transportation. Several research gaps were identified, but the most frequent across countries was the need to study active transportation to destinations other than school (Table 5).

## 4. Discussion

Our results suggest that about half of children and adolescents use active modes of transportation to get to and from places, mainly to and/or from school. However, a pooled estimate of the global prevalence of active transportation cannot be calculated from the Global Matrix 3.0 data for reasons that will be discussed below. Despite the clear gradient in average grades according to HDI that has been discussed in previous publications [141,142,143], our results show variability within HDI groups and the LPA allowed us to examine the clustering of this sample of countries according to three variables of interest (active transportation grades, level of development and income inequality).

### 4.1. Comparability of Data

There was wide variability between countries in the prevalence of active transportation, and high involvement in this behaviour was reported across countries with very different socioeconomic contexts (e.g., Japan, Zimbabwe, Nepal, Denmark and Finland). However, the data reported by the countries presented in Table 3 show important methodological differences that should be accounted for when comparing the prevalence of active transportation between countries. One of the issues that can affect the comparability of data is the difference in the frequency of use of active transportation reported by the countries. Depending on the cut-point used to define children as active travelers, the prevalence will vary widely, and the use of active transportation can be overestimated or underestimated. Similarly, the prevalence may vary depending on the direction of active transportation assessed since different modes can be used to go to and from school. As observed in previous comparisons of surveillance systems measuring active transportation, the prevalence of active transportation varies greatly according to the construct assessed [144]. In the group of countries included in this analysis, the frequencies reported varied from daily to at least twice per week. Even when the source of information was the same survey (e.g., the GSHS across countries), different frequencies were reported [136,145,146,147]. Regarding the construct assessed, the destination for active transportation is also relevant. Despite the broad definition of active transportation in the Global Matrix 3.0 benchmarks [14], most of the evidence available on active transportation in children is focused on the journeys to and from school, as observed in this analysis and in previous literature [148]. Only Ecuador and the United States reported the use of active transportation to other destinations, which could suggest an underestimation of the involvement in active transportation in other countries since trips to places such as parks and other people’s homes are also relevant opportunities to engage in this behaviour [149]. These findings point to a need for the development of harmonized and contextualized measurements. Our results are consistent with the findings reported by Herrador-Colmenero et al. in a systematic review, in which the formulation of a standardized question is proposed to overcome the heterogeneity in measures to assess active transportation [150]. Based on these insights, initiatives like the Global Matrix and organizations like the AHKGA can contribute to the improvement of surveillance systems for the evaluation of active transportation among children.

The Global Matrix initiative aims to better understand the global variation of certain physical activity indicators [14]. Specifically, active transportation is one of the most strategic indicators in the Global Matrix 3.0 to contribute to this aim, due to the low amount of INC grades, and the good dispersion of grades across countries [14]. However, the availability of transportation-relevant contextual variables at the country level to understand these variations was limited. Therefore, the LPA provides an exploratory approach to identify subgroups that share similar patterns of variables [20,151], and provides a unique opportunity to identify the ways in which countries in the Global Matrix 3.0 cluster, according to the grades for active transportation and contextual variables. The identified profiles can be useful for the discussion of the different contexts in which active transportation needs to be maintained or increased. A description of the three profiles is provided below.

### 4.2. Country Profiles for Active Transportation and Sociodemographic Variables

Profile 1 included mainly countries with a very high HDI and low income inequality, mostly with a reported prevalence of active transportation under 50%. Mainly, countries from North America, Europe and Oceania were grouped in this profile. While the countries with the lowest prevalence of active transportation were classified in this group (Chile, the United States and Canada), it also included some countries with non-negligible prevalence of active transportation such as the Netherlands, Belgium and the Czech Republic. This means that although all of these countries have a similar development level, there are other relevant factors influencing active travel among children. First, some of these are countries where long distances between destinations and the perceived convenience of driving may undermine opportunities for active travel [102,152,153,154]. Second, urban planning and policies that have prioritized people instead of cars, as well as supportive infrastructure have made active modes a convenient and safe alternative to commute [155,156]. Interventions in countries under this profile should aim to increase active transportation addressing the issues of distance and convenience, attempting to discourage the use of motorized vehicles for short trips, and trying to shift the social norms to consider active modes the default option for commuting as it occurs in many European countries. A useful example among the policies reported in the Report Cards is the National Cycling Policy from Sweden, which aims to prioritize cycling in the community and municipalities planning [123].

Profile 2 grouped mostly countries with high prevalence of active transportation, low to medium HDI and higher income inequalities. In most of these countries, access to motorized vehicles is limited, and active travel is happening despite multiple safety concerns [157,158] and the lack of supportive infrastructure [143]. Therefore, for many families, active transportation is likely to reflect necessity rather than choice [159]. Also, many of the countries in this group report important differences between children from rural and urban areas [117,120,145]. As suggested by a previous systematic review on active transportation in Africa, these differences could be indicative of the physical activity transition that these countries are experiencing [157,160]. In this context, for the countries classified in this profile, preserving active travel while providing improved safety and infrastructure conditions should be a priority. It is important to design strategies to avoid the unintended consequences that economic growth can have on the mode of transport for children and adolescents. A good example of the approaches needed in countries under this profile is the Non-Motorized Transport Policy from Lagos, Nigeria. This policy aims to prioritize active modes of transportation over motorized options, communicating the benefits and importance of active transportation, as well as improving safety conditions for students using active modes to go to school [116].

Profile 3 had more variability in terms of HDI and income inequality, however, the relatively high prevalence of active transportation was a main feature in common between this group of countries. Some of the most successful countries in active transportation are grouped under this profile. However, the conditions in which it is happening are very different. There are countries such as Finland, Denmark, Japan, South Korea and Hong Kong where the use of active modes is supported by the design of compact cities, school siting policies that ensure that children attend to schools located at a walkable distance from home, and supportive infrastructure and regulations [103,104,108,141,155,161]. These factors have made walking and cycling safe options for the daily commuting. Conversely, there are countries like Colombia, Brazil, Mexico, Venezuela and South Africa, where active transportation is prevalent despite safety concerns, the lack of supportive infrastructure and regulations and is likely to be a necessity-driven behaviour [52,60,61,162,163,164,165]. Similarly to profile 2, almost half of the countries in this profile have a relatively high Gini coefficient. However, this profile also includes countries with very low inequality, such as Finland and Denmark. Income inequality has been previously documented as a negative correlate of physical activity and organized sports involvement [14,19]. Notwithstanding, the high prevalence of active transportation in both equal and unequal societies are consistent with literature that suggest that active transportation modes could be an opportunity to bridge the inequities in transportation [18] as well as in other domains of physical activity. Due to the diversity of contexts found in this profile, different approaches are needed to promote or maintain active travel. School siting policies that take into account the proximity between schools and children’s homes, like those implemented in Japan and Hong Kong [103,104,107], can be useful for growing cities. Also, multi-component strategies, such as the Bike to school program in Colombia are a good reference for countries that aim to provide access, skills, and support to bike to school in safe conditions [110]. Furthermore, Ciclovias or Open Streets programs are a good model for countries where active transportation to school is already prevalent and aim to increase walking and cycling to other destinations in the leisure time [112,166].

Regarding the strategies to improve active transportation, it is concerning to find that major correlates of active transportation such as distance and the perceived convenience of driving are not mentioned among the strategies proposed by the Report Card teams. Future versions of the Report Cards, as tools to communicate evidence to stakeholders, should take these important factors into consideration in order to advocate for active transportation addressing its most important drivers.

Our results can contribute to the call for measures of conditions related to all children wellbeing made by a recent commission sponsored by the WHO, UNICEF and The Lancet. This commission identified that inequities and climate change are undermining children’s right to a healthy environment in both, the poorest and wealthiest countries [167]. Given that the transportation sector accounts for almost 25% of global greenhouse gas emissions [168], local, regional, and national policymakers and practitioners should implement interventions that support children’s active transportation in all socioeconomic contexts.

### 4.3. Strengths and Limitations of the Study

Strengths of this study include the availability of active transportation data from 47 countries from all continents, and the harmonized selection of the best available evidence in each country. Our analyses contributed with a diverse context perspective to the emerging evidence on international comparisons of active transportation, which has focused on specific groups of countries in previous studies [169,170]. Although most countries reported nationally representative data on active transportation, in some countries, the best available evidence consisted of local data. The main limitations of the study were the diversity in the quality of the data reported, and the broad benchmark proposed for active transportation in the Global Matrix 3.0, which led to variations in the definition of active transportation across countries. The important amount of missing data in the Community and Environment indicator (26%) and the heterogeneity of the data reported across countries did not allow to include it as a variable of interest in the LPA, despite its relevance for active transportation. For example, including data on average distances for active transportation by country in future studies could strengthen the model and enrich the profiling of countries as distance is one of the most consistent predictors of active transportation. Also, since we analyzed aggregated data at the country level, a sample size of 47 is small and has limited power for the LPA. This could partly explain the heterogeneity observed in the profiles, mainly in profile 1. Regarding the policies and practices reviewed, there was also heterogeneity in the information reported across countries. Future versions of the Global Matrix can strengthen the guidance on desirable information to report in this regard, such as the inclusion of active transportation to school in National Education Acts or their equivalents in each country. The sample included in this study represents approximately 25% of the total countries in the world. The inclusion of a larger sample of countries in future studies could provide a clearer picture of profiles according to active transportation and sociodemographic variables.

## 5. Conclusions

This work allowed for a deeper exploration of the active transportation information reported by all the countries participating in the Global Matrix 3.0. Based on our findings, we identified the need to standardize definitions of active transportation to be able to make more meaningful comparisons. The LPA conducted allows for the inference that countries belonging to a specific profile have a greater probability of sharing certain characteristics among them compared to the countries belonging to other profiles. Given the variation by geographic region and even HDI, this approach is useful for identification of more meaningful groupings that can facilitate the cross-fertilization of efforts to promote active transportation, and therefore, to “power the movement to get kids moving”, as is intended by the Global Matrix initiative [171]. The Active Healthy Kids Global Alliance can contribute to improving active travel surveillance providing guidance to countries involved in future versions of the Global Matrix. A more comprehensive approach to active transportation surveillance that considers duration, distance, frequency, direction, other destinations than school and the contribution of active transportation to school to overall active transportation, could improve the understanding of this behaviour and its potential to increase overall physical activity.

## Figures and Tables

**Figure 1 ijerph-17-05997-f001:**
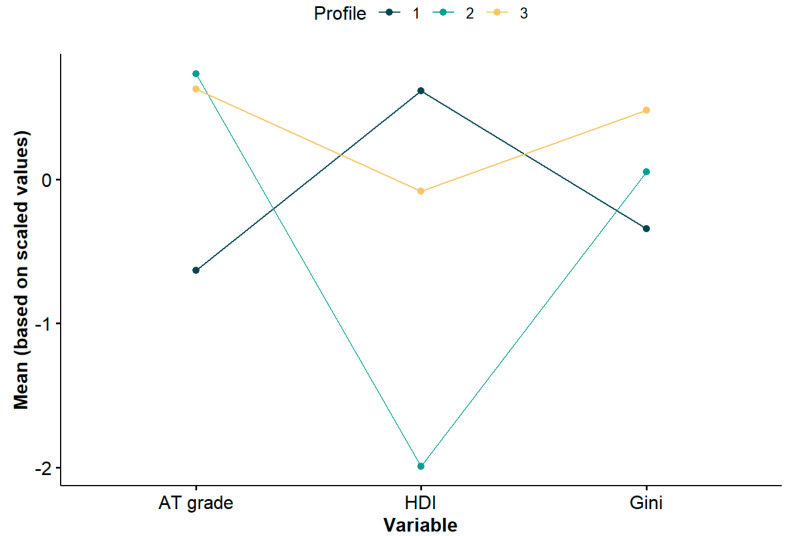
Country profiles for active transportation and sociodemographic variables of countries in the Global Matrix 3.0. The range of values for the active transportation grade, Human Development Index and the Gini index varied notably between variables, therefore they were converted to z-scores to be expressed in the same range of values and to ease their graphic depiction.

**Table 1 ijerph-17-05997-t001:** Global Matrix 3.0 grading rubric.

Grade	Interpretation ^a^	Numerical Equivalents ^b^
**A+**	94–100%	15
**A**	We are succeeding with a large majority of children and youth (87–93%)	14
**A−**	80–86%	13
**B+**	74–79%	12
**B**	We are succeeding with well over half of children and youth (67–73%)	11
**B−**	60–66%	10
**C+**	54–59%	9
**C**	We are succeeding with about half of children and youth (47–53%)	8
**C−**	40–46%	7
**D+**	34–39%	6
**D**	We are succeeding with less than half but some children and youth (27–33%)	5
**D−**	20–26%	4
**F**	We are succeeding with very few children and youth (<20%)	2
**INC ^c^**	Incomplete—insufficient or inadequate information to assign a grade	

^a^ For this article, the interpretation corresponds to the percentage of children and youth who use active transportation to get to and from places (e.g., school, park, mall, friend’s house). ^b^ Letter grades were converted to numerical equivalents for analyses purposes. ^c^ INC: incomplete

**Table 2 ijerph-17-05997-t002:** Active transportation grades and sociodemographic variables of the 49 countries participating in the Global Matrix 3.0.

Country	Active Transport Grade	Active Transport Numerical Grade	Human Development Index (HDI) ^a^	HDI Classification	Gini Index ^b^
Australia	D+	6	0.939	Very high	34.7
Bangladesh	C−	7	0.579	Low to medium	32.4
Belgium (Flanders)	C+	9	0.896	Very high	27.7
Botswana	C	8	0.698	Low to medium	60.5
Brazil	C	8	0.754	High	51.3
Bulgaria	B−	10	0.794	High	37.4
Canada	D−	4	0.920	Very high	34.0
Chile	F	2	0.847	Very high	47.7
China	C+	9	0.738	High	42.2
Colombia	B	11	0.727	High	50.8
Czech Republic	C+	9	0.878	Very high	25.9
Denmark	B+	12	0.925	Very high	28.2
Ecuador	C−	7	0.739	High	45.0
England	C−	7	0.909	Very high	33.2
Estonia	D	5	0.865	Very high	32.7
Ethiopia	C	8	0.448	Low to medium	39.1
Finland	B+	12	0.895	Very high	27.1
France	C−	7	0.897	Very high	32.7
Germany	C−	7	0.926	Very high	31.7
Ghana	C+	9	0.579	Low to medium	42.4
Guernsey Channel Islands	D	5	0.975	Very high	40.0
Hong Kong	B+	12	0.917	Very high	N/A
India	B−	10	0.624	Low to medium	35.1
Japan	A−	13	0.903	Very high	32.1
Jersey	D+	6	0.985	Very high	41.0
Lebanon	D	5	0.763	High	31.8
Lithuania	C−	7	0.848	Very high	37.4
Mexico	C+	9	0.762	High	43.4
Nepal	A−	13	0.558	Low to medium	32.8
Netherlands	B−	10	0.924	Very high	29.3
New Zealand	C−	7	0.915	Very high	N/A
Nigeria	B	11	0.527	Low to medium	43.0
Poland	C	8	0.855	Very high	31.8
Portugal	C−	7	0.843	Very high	35.5
Qatar	N/A	N/A	0.856	Very high	N/A
Scotland	C	8	0.909	Very high	33.2
Slovenia	C	8	0.890	Very high	25.4
South Africa	C	8	0.666	Low to medium	63.0
South Korea	B+	12	0.901	Very high	31.6
Spain	B−	10	0.884	Very high	36.2
Sweden	C	8	0.913	Very high	29.2
Taiwan	C−	7	0.885	Very high	33.6
Thailand	C	8	0.740	High	37.8
United Arab Emirates	INC	N/A	0.840	Very high	N/A
United States	D−	4	0.920	Very high	41.5
Uruguay	C	8	0.795	High	39.7
Venezuela	B−	10	0.767	High	46.9
Wales	D+	6	0.909	Very high	33.2
Zimbabwe	A−	13	0.516	Low to medium	43.2
Global average	C	8.29	NA	NA	NA
Low to medium HDI countries	C+	9.67	NA	NA	NA
High HDI countries	C	8.5	NA	NA	NA
Very high HDI counties	C−	7.78	NA	NA	NA

^a^ Data at the national level from the United Nations Development Programme [16]. ^b^ Data at the national level from the World Bank [17]. Abbreviations: HDI, Human Development Index; INC, Incomplete, N/A, Not available; NA, Not Applicable.

**Table 3 ijerph-17-05997-t003:** Rationale for grades and information reported on active transportation by 49 countries involved in the Global Matrix 3.0.

Grade	Country	Rationale	Gender	Age	Destination and Direction	Frequency	Source of Information and Year	Profile
A−	Japan	86% of students used active transportation to school from home.	Not reported	6–17 years	To school	On a regular basis	The National Sports-Life Survey of Young People (SSF), 2015 [25]	3
A−	Nepal	86% of children and youth of 15–20 years used active transportation to get to and from places.	Not reported	15–20 years	Not specified	Not clear	Physical Activity Level and Associated Factors Among Higher Secondary School Students in Banke, Nepal: A Cross-Sectional Study, 2013 [26]	2
A−	Zimbabwe	Over 80% of children and adolescents used active transport to and from school, with variation between provinces as well as between rural and urban areas.	82% of girls and 79% of boys engaged in active transport to and from school	8–16 years	To and from School	Not clear	The Zimbabwe Baseline [27] and Global school-based health survey (GSHS) Zimbabwe 2003 [28]	2
B+	Denmark	78% of children and adolescents reported cycling, walking, or using children’s scooters as transport (e.g., to school) at least two times per week.	Not reported	7–15 years	To school	At least two times per week	Danish Sports Habits Study 2016 [29]	3
B+	Finland	77% of children and adolescents actively commuted to school, on foot or by bike.	9 years old 79% of boys, 81% of girls 11 years old 85% of boys, 81% of girls 13 years old 80% of boys, 77% of girls 15 years old 59% of boys, 63% of girls	9–15 years	To school	Not clear	National Physical Activity Behaviour Study for Children and Adolescents 2016 (LIITU) [30]	3
B+	Hong Kong	80% of the adolescent males and 77% of the adolescent females actively travelled to school at least once per week. 52% of primary school children used active travel to/from school at least 5 times per week.	80% adolescent males and 77% adolescent females	Primary and secondary	To and from school	At least 5 times per week and at least once per week	Understanding Children’s Activity and Nutrition (UCAN) study, 2011–2012 [31]	3
B+	South Korea	79.4% of children and adolescents reported walking or cycling to/from places, with an average duration of 39 min per day.	84.3% of boys and 73.8% of girls took active modes of transport	12–17 years	Not specified	Not clear	Korea National Health and Nutrition Examination Survey, 2016 [32]	3
B	Colombia	71.7% of children and adolescents in Colombia reported walking or biking as their main mode of transport to or from school in the previous week.	Not reported	6–17 years	To and from school	Main mode during the last 7 days	National Survey of Nutrition (ENSIN) 2015 [33]	3
B	Nigeria	The majority (61% to 80%) of Nigerian children and adolescents engage in some form of active transportation, mostly walking to and from school.	Not reported	5–13 years	To and from school	Not clear	2 different studies on rural and urban populations in Nigeria, conducted in 2011 [34] and 2013 [35]	2
B−	Bulgaria	64% of children and youth reported walking, biking, or skating, etc. to go to school and back.	Not reported	Not specified	To and from school	Not clear	Bulgarian Active Kids survey, 2016 [36]	3
B−	India	Approximately 65% of children/adolescents (weighted average) reported walking or cycling to school on a regular basis.	Not reported	5–17 years	To school	On a regular basis	7 different studies at the national and local level conducted between 2005 and 2018 [12,13,37,38,39,40,41]	2
B−	Netherlands	90% of the adolescents commute actively to school. 36% of the children commute actively to school	Not reported	Not specified	To school	At least three days per week	Lifestyle monitor National Survey, 2017 [42]	1
B−	Spain	55% and 56.9% of children between 6 and 9 years old walked to and from school, respectively. In Catalonia, 61.3% of children between 3 to 14 years old walked to and from school.	Not reported	3–14 years	To and from school	Not clear	Food, Physical Activity, Child developmentand Obesity study (ALADINO) 2011 [43], and the Catalan Health Survey (ESCA) 2016 [44]	3
B−	Venezuela	63% of adolescents might walk at least 10 min to move from one place to another.	Not reported	Not specified	Not specified	Not clear	Venezuelan Study of Nutrition and Health [45]	3
C+	Belgium (Flanders)	55.5% of parents of 6-to 9-year old children reported that their child uses active transportation, and 58.9% of 10- to 17-year-olds adolescents reported to mainly use active transportation to travel to school.	Not reported	6 to 9 and 10 to 17	To school	Not clear	Belgian National Food Consumption Survey 2014 [46]	1
C+	China	56.3% of Chinese children (aged 6–18 years) reported going to and from school by walk or bicycle.	Not reported	9 to 17 years	To and from school	Daily	Physical Activity and Fitness in China—The Youth Study (PAFCTYS), 2016 [47]	3
C+	Czech Republic	On average, 57% (weighted mean: 59%) of children and adolescents reported using active transport to get to and from school.	Not reported	9–17 years	To and from school	Not clear	Health Behaviour in School-aged Children Study (HBSC)2006, 2010, and 2014 [48] and International Physical Activity and Environment Network Study (IPEN), 2013–2015 [49]	1
C+	Ghana	About 54% of children and youth especially those in the rural areas walk to school and back home covering about 2 km.	Not reported	Not specified	To and from school	Not clear	Not specified	2
C+	Mexico	54.8% of children 3 years and older walked to school and 1.5% rode bicycles. 69% of 10–14-year-olds walked or rode a bicycle to school.	Not reported	3 years and older	To school	Not clear	Intercensal Survey of the National Institute of Statistics and Geography (INEGI), 2015 [50] and the National Health and Nutrition Survey 2016 (ENSANUT) [51]	3
C	Botswana	49% of 13–15-year-olds walked or rode a bike to and from school at least one day during the past 7 days.	Not reported	13–15	To and from school	At least one of the last 7 days	2005 Botswana School-based Student Health Survey (GSHS) [13]	3
C	Brazil	55.0% of children and youth in Brazil used active transportation to get to and from school.	Not reported	6 to 21	To and from school	Not clear	18 different national and regional studies conducted between 2008 and 2017 [52]	3
C	Ethiopia	Approximately 48% of children and youth (31% in urban and 65% rural) walked to and from school.	Not reported	Not specified	To and from school	Not clear	Experts’ opinion	2
C	Poland	47.4% of 10- to 17-year-olds reported walking to school and 52.3% walking from school. 5.5% and 5.2% travel to and from school by bicycle, respectively. 41% and 5% of lower-secondary students walk and cycle to school, respectively. While 36% and 3% of upper-secondary school students walk and cycle to school, respectively.	Not reported	10–19 years	To and from school	Not clear	Study of Physical Activity of School Children Aged 9–17 by the Institute of Mother and Child, 2013 [53] and the All-Poland survey of physical activity and sedentary lifestyles for middle school, high school and university students, 2011 [54]	1
C	Scotland	51% and 52% of school age children and adolescents, respectively, actively commuted to school (walking, cycling, skating or using scooter).	Not reported	4–18 years	To school	Not clear	Hands up Scotland Survey (HUS) 2016 [55], Transport and Travel in Scotland (TATiS) 2016 [56]	1
C	Slovenia	Almost 49% of children commute actively to and from school and additional 12% commute actively from school only.	52% of boys and 50% of girls commute actively to school	5–15 years	To and from school	Not clear	Analysis of Children’s Development in Slovenia study (ACDSi) 2013–16 [57,58]	1
C	South Africa	63% of school-aged children walk to school. 81% of children and adolescents in Cape Town walk to school without adult supervision in low-income settings, and 61% of parents reported concerns about their children safety.	Not reported	6–15 years	To school	Not clear	General Household Survey, 2013 [59] and two local studies conducted in Cape Town, 2016 [60,61]	3
C	Sweden	48% and 57% of children and adolescents used active transportation to and from school in the winter and summer months, respectively.	Not reported	6–15 years	To and from school	Not clear	Children’s Routes to School Survey 2015–16 [62]	1
C	Thailand	53.4% children and adolescents used active transportation (walking, cycling, using a wheelchair, in-line skating or skateboarding) to get to and from places.	54.7% of girls and 52.4% of boys used active transportation	6–17 years	Not specified	Not clear	Thailand Physical Activity ChildrenSurvey (TPACS) 2015 [63]	3
C	Uruguay	51.2% of adolescents between 13 and 15 years old went to the school walking or bicycling 4 or more days per week.	Not reported	13–15 years	To school	4 or more days per week	Global School-Based Student Health Survey (GSHS) 2012 [64]	3
C−	Bangladesh	41.1% students aged 13–17 years used active transport to commute to or from school at all seven days prior to the survey.	Not reported	13–17 years	To and from school	Last 7 days	Bangladesh School-based Student Health Survey (GSHS), 2014 [65]	2
C−	Ecuador	42.7% of 5–17 years-old children reported going to school or work by foot or bike.	Boys: 42% Girls: 41%	5–17 years	To school, work or other destinations	Not clear	Not specified	3
C−	England	On average, 42.5% of children and adolescents used active modes of transport to school everyday.	Not reported	5–16 years	To school	Every day	National Travel Survey 2016 [66], Health Survey for England 2015 [67] and Walking and Cycling Statistics 2016 [68]	1
C−	France	44% of the 3–10 years old and 43% of the 11–14 years old used active transportation to go to school according to the National Study of Individual Nutritional Consumption 2014–2015. And 41% of the 6–10-year-olds reported using active transportation to school according to the Health Study of the Environment, Biosurveillance, Physical Activity, and Nutrition 2015.	Not reported	3–14 years	To school	Not clear	National Study of Individual Nutritional Consumption (INCA) 2014–2015 [69] and the Health Study of the Environment, Biosurveillance, Physical Activity and Nutrition (ESTEBAN) 2015 [70]	1
C−	Germany	Approximately 40% of the children and adolescents commute actively to school.	Not reported	Not specified	To school	Not clear	Not specified	1
C−	Lithuania	45% of 7–8 aged children used active transport to school and 57.9% used active transport from school to home. 84% of children and adolescents (11–13 year) walked to/from school. 12% of youths and adolescents (15–24 year) reported to engage regularly in cycling from one point to another.	Not reported	7–24 years	To and from school	Not clear	4 different studies and one special report Eurobarometer on Sport and Physical Activity, between 2012 and 2017 [71,72,73,74,75]	1
C−	New Zealand	45% of children and adolescents aged 5–14 years usually used active transport to school according to the NZ Health Survey, 43% of children and youth aged 5–17-year-olds usually used active transport to and from school according to the Active NZ Survey. 30% of children aged 5–12 years and 31% of adolescents aged 13–17 years used active transport to school according to the Health and Lifestyles Survey and the NZ Household Travel Survey, respectively. 24% of 6-year-olds in a longitudinal cohort study usually used active transport to get to and from school.	Not reported	5–17 years	To and from school	Not clear	New Zealand Health Survey 2016/2017 [76], Active NZ Survey 2018 [77], Health and Lifestyles Survey 2016 [78], NZ Household Travel Survey 2015–2018 [79], and the Growing Up in New Zealand study 2010 [80]	1
C−	Portugal	A study with urban school-aged children showed that 45% of participants commuted actively to and from school. Another study in the countryside region found that 30% of the participants (aged 7 to 8 years) commuted either by foot or cycling on a regular basis during school days (ARSA 2012).	Not reported	7–8 years and 15–24 years	To and from school	Not clear	A study in public schools from the Porto area [81] and the Health Study of the Child Population of the Alentejo Region, 2012 [82]	1
C−	Taiwan	33–46% of children and adolescents reported walking or cycling to school most of the days.	Not reported	7–18 years	To school	Most of the days	Student Participation in Physical Activity Survey, 2015 [83] and Health Behaviour Survey in Junior High School Students, 2016 [84]	1
D+	Australia	National data shows that 43% of 12–17 year-olds usually travel to/from school using active transport, other state and regional studies shows that 37% of primary students and 36% of secondary students use active transport as their usual mode to get to school.	Not reported	12–17 years	To school	Usual mode—each week (Usual defined as at least 5 trips out of 10 or on at least 2.5 school days)	National Secondary Students Diet and Activity Survey 2012–2013 [85], ACT Year 6 Physical Activity and Nutrition Survey 2015 [86], Child Population Health Survey [87], Queensland Child Preventive Health Survey 2018 [88], NSW School Physical Activity and Nutrition Survey 2015 [89], Victorian Child Health and Wellbeing survey 2016 [90]	1
D+	Jersey	37% of 10–15-year-olds traveled to school by active modes.	Not reported	10–15 years	To school	Not clear	Health Related Behaviour Questionnaire 2014 [91]	1
D+	Wales	44% primary school children and 34% secondary school pupils traveled actively to school. In another survey, 33.8% and 36.1% of children and young people aged 11–16 years walked/cycled to and from school, respectively.	Not reported	11–16 years	To and from school	Not clear	National Survey for Wales (2016–17) [92] and The Health Behaviour of School-aged Children (HBSC)/School Health Research Network (SHRN) Survey 2017 [93]	1
D	Estonia	The percentage of use of active transport varied between 36–56%. Specifically, 35% of children walked to school and back home, while 14% of children rode a bike to go to school. The grading process took into account the number of subjects, age range and used methodology of different studies.	Not reported	7–17 years	To and from school	Not clear	Children’s Physical Activity Study 2015 and Schools in Motion Survey 2018 [94]	1
D	Guernsey Channel Islands	On average, 31% of children and adolescents reported active travel (walking, bicycle or scooter) to school on the day of the survey (43% of primary school pupils and 25% of secondary pupils.	Not reported	Primary and secondary (grades 6, 8 and 10)	To school	On the day of the survey	Guernsey Young People Survey 2016–2017 [95]	1
D	Lebanon	36.8% of Lebanese adolescents between the ages of 13 and 18 reported walking or biking to school.	Not reported	13–17 years	To school	Not clear	Global School-Based Student Health Survey 2016 (GSHS) [96]	1
D−	Canada	21% of 5- to 19-year-olds in Canada typically use active modes of transportation (e.g., walk, bike), and 16% use a combination of active and inactive modes of transportation to travel to and from school (2014–16 CANPLAY, CFLRI).	Not reported	5 to 19	To and from school	Typical use	Kids CANPLAY 2014–2016 [97]	1
D−	United States	38% of adolescents walked or used a bicycle for at least 10 min continuously once or more in a typical week to get to and from places, and 23% of youth actively commuted 5–7 days per week.	45% of boys and 32% of girls reported any active transportation in a typical week	12–19 years	To and from multiple places	5 to 7 days on a typical week	National Health and Nutrition Examination Survey (NHANES 2015–16) [98]	1
F	Chile	15% of children and youth (weighted average) rode a bike or walked to and from school (ranging from 0.29% to 32.2% in different samples and regions).	Not reported	Not specified	To and from school	Not clear	National Survey of Quality of life 2015–2016 (ENCAVI) [99], Survey of Urban Quality of Life Perception 2015 (EPCVU) [100], a cross-sectional study of seventh grade students in the Maule region 2014 [101], and a cross-sectional study in Valparaiso 2017 [102]	1
INC	United Arab Emirates	There was no current data available to grade this indicator.	NA	NA	NA	NA	NA	1
N/A	Qatar	Active transportation indicator is excluded from the report according to stakeholders’ group recommendation. This indicator is still not applicable in Qatar due to unsafe roads and the hot climate during most times of the year.	N/A	N/A	N/A	N/A	N/A	1

Abbreviations: INC, incomplete; NA, not applicable.

**Table 4 ijerph-17-05997-t004:** Descriptive statistics of the latent variables by country profile.

Profile (% of Countries)	Active Transportation Grade				Human Development Index				Gini Index			
	Mean	SD	Min	Max	Mean	SD	Min	Max	Mean	SD	Min	Max
1 *n* = 25 (51%)	6.08	2.55	0.00	10.00	0.89	0.05	0.763	0.985	33.78	5.29	25.44	47.70
2 *n* = 7 (14.3%)	10.14	2.34	7.00	13.00	0.55	0.06	0.448	0.624	38.29	4.81	32.40	43.20
3 *n* = 17 (34.7%)	9.82	1.88	7.00	13.00	0.80	0.09	0.666	0.925	42.08	10.58	27.10	63.0

**Table 5 ijerph-17-05997-t005:** Policies and practices, strategies to improve the grade and research gaps in active transportation identified in the Global Matrix 3.0.

Grade	Country	Profile	Policies/Practices	How to Improve the Grade	Gaps
A−	Japan	3	Since 1953 Japan has a “walking to school practice” resulting from the implementation of the article 49 of the School Education Act, which regulates the siting of public schools in urban areas of Japan. This article establishes that the commuting distances are 4 km for elementary schools and 6 km for junior high schools. Based on these, the boards of education must ensure that children attend to schools located within those distances to allow children to walk to school [14,103,104].	Not reported	Research on active transportation to destinations other than schools (e.g., going shopping, going to the park, sports clubs or cram schools) [104].
A−	Zimbabwe	2	Not reported	1. Through public health messages to highlight the benefits of active transportation and reduce the prestige/status symbol associated with motorized transportation. 2. Implementing policies that encourage and provide safe and walkable neighborhoods and bike lanes, etc. [105].	There is a need of data reporting the time invested in active transportation and distance to and from school, as well as research data on the correlates of active transportation, and more recent data is required [105].
B+	Finland	3	Not reported	Not reported	There is no comparable published data available about active school commutes for upper secondary students when the distance between home and school is less than 5 km. More information is needed about active transportation to other destinations [106].
B+	Hong Kong	3	The high density of Hong Kong could be one of the factors facilitating active transportation to school. Since most districts in Hong Kong are highly self-contained, children can attend schools located at walkable distance from their home [107,108].	1. Encouraging active travel to destinations other than school may provide additional health benefits for children and adolescents. 2. Promoting cycling to and from school and other destinations in districts with a bicycle track [107,108].	Data about active transportation to destinations other than school, as well as the relationship between active transportation, physical activity and health-related outcomes. Also, data on the duration of active travel trips is required [107,108].
B	Colombia	3	In Bogota, the capital city of Colombia, the program “Bike to school” is implemented in public schools to promote cycling as a sustainable mode of transportation to school and other destinations in the city. The program was created in order to address the barriers to access to education and to decrease the dropout rates. Bike to school program includes the following strategies: (a) bicycle loan, (b) workshops on skills and abilities to ride a bicycle, (c) basic mechanics and road safety education, (d) participatory mapping of safe routes, (e) daily trips from a meeting point to school with adult supervision, and (f) extracurricular activities to develop responsible behaviours in the roads and to visit other destinations of interest in the city [109,110]. Another promising practice to encourage walking and cycling, but with recreational purposes, are the Open Streets programs, or Ciclovias. Colombia currently has 67 of these programs that close main roads to motorized vehicles and open them for leisure activities on Sundays and holidays [109,111]. Walking and cycling are the main activities performed by children who attend Ciclovia in Bogota [112]. Also, Colombia has a specific law to support the use of bicycles as the main mode of transport at the national level (Law 1811 of 2016). This law establishes the responsibility of public transportation systems to allow multi-modal trips through the provision of bike-supporting infrastructure, and encourages schools to implement programs to promote cycling [113].	1. Improving safety conditions and infrastructure to keep promoting and maintaining active transportation as a desirable behaviour since early ages [109].	Not reported
B	Nigeria	2	The National Transport Policy in Nigeria is under review with the aim to strengthen the inclusion of non-motorized transport infrastructure and to create better non-motorized transport options for urban residents. This review is the result of a workshop on streets design led by the Federal Ministry of Transport in 2017 and is a good example of the concerted efforts to improve the conditions for active transportation in Nigeria [114,115]. Another example is the Non-Motorized Transport Policy developed in Lagos, which aims to prioritize walking, cycling and public transportation as the main modes of transport [114]. This policy specifically addresses active transportation to school through two strategies: (a) public awareness through the creation of a curriculum about road safety and benefits of active transportation for primary and secondary school students. And (b) regulations that include the creation of route plans for students to go to school, and the implementation of access and safety measures such as speed limits, traffic calming infrastructure and school zone signaling [116].	Not reported	Not reported
C+	Ghana	2	The Community Day Senior High Schools, built in various districts in Ghana, seem to be encouraging active transportation to school. The students who attend to this schools usually walk to and from school every day, some of them covering more than two kilometers [117].	Not reported	Not reported
C+	Mexico	3	Not reported	1. Promoting active transportation among Mexican children and adolescents. 2. Communities and governments should provide appropriate safety conditions on streets, sidewalks and neighborhoods to promote walking and cycling among children and adolescents [118].	Data on all age groups and stratified by age group and sex is desirable for future surveys [118].
C	Brazil	3	Not reported	Local authorities should be encouraged to create a monitoring system to generate standardized and detailed reports on active transportation to school to support planning and evaluation of public policies [119].	Data on time invested in active transportation, the distance to the school and other environmental and mobility-related factors such as bike paths, traffic and conditions of the city is lacking [52].
C	Ethiopia	2	Not reported	1. Building sidewalks to encourage active transportation in all cities in Ethiopia. 2. Encouraging and supporting children and adolescents to travel to and from school through active transportation [120].	Active transportation specific studies in Ethiopia are required [120].
C	Scotland	1	Not reported	Not reported	No data available on active commuting to and from places other than school [121].
C	Sweden	1	A national cycling strategy has been adopted in Sweden to improve safety and increase cycling [122]. The strategy aims to increase cycling through five action lines: (a) creating more bicycle-friendly municipalities, (b) focusing on various types of cyclists (where children are highlighted as a population of interest), (c) giving higher priority to bicycle traffic in community planning, (d) building more functional and user-friendly cycling infrastructure and (e) strengthening research an innovation on cycling [123].	Not reported	Not reported
C	Uruguay	3	Not reported	Creating policies to encourage the creation of cycle lanes and safe sidewalks.	Data on active transportation in a wider age range and to locations other than school.
C−	Ecuador	3	Not reported	Reinforcing programs aiming to promote active transportation [124].	Not reported
C−	France	1	Not reported	Not reported	Research is needed on the characteristics of active transportation of children and adolescents (frequency, mode, distance covered) and on the potential barriers to this in order to develop effective promotion program [125].
C−	Lithuania	1	Not reported	1. Promoting and facilitating safe active transport to get to school and other destinations. 2. Prioritizing active transportation promotion as a key factor at schools and communities. 3. Involving parents, schools, community and policy makers in the promotion of active transportation [71].	1. Research on the prevalence and trends of active transport in Lithuania, considering the most popular modes of active transportation used to get to/from different points or destinations (e.g., parks, shops, sport fields) among children and adolescents as well as studying the role of active transport in achieving recommended levels of physical activity. 2. Research on health and social benefits of active transportation is needed. 3. Evaluating the impact of the cycling paths and interventions at the school, community and municipality levels. 4. Examining the potential moderators and mediators of active transport behaviour change to help refine interventions [71].
C−	New Zealand	1	Not reported.	Strategies to encourage active travel to school are needed, especially for girls, younger children, and older adolescents [126]. This strategies should have a multi-sectoral y culturally appropriate approach, including urban planning, initiatives a the school and community level, social marketing campaigns and family support [127].	Nationally representative data on active transportation to school and other destinations that is comparable between countries and across time is desirable [127].
C−	Taiwan	1	Not reported	Local governments and schools should work together to create a safe and convenient environment for active transportation [128].	Research on the contribution of active transportation to overall physical activity in children and adolescents, and about motivations and barriers for active transportation is needed [128].
D+	Australia	1	The Australian Capital Territory has implemented the Ride or Walk to School program since 2012 aiming to build the capacity of primary schools to support and promote active travel to and from school. The program was designed with a participatory approach including students and different stakeholders. The strategies of the program include: (a) resources for teachers and students, (b) provision of bikes and helmets, (c) safe routes maps, (d) workshops to increase skills, and (e) four annual active travel events. This program was expanded to high schools since 2016 [129]. In Western Australia, the Department of Transport implemented the program “Your Move Schools”. This is a community-focused program that promotes active and sustainable transportation providing: (a) teaching resources, (b) expert advice and (c) access to funding (up to $5000 AUD) to promote active transportation through bike education workshops, wayfinding, bike supporting infrastructure like bike shelters, bike repair stations, bike skills tracks and bicycle parkings [129].	1. Encouraging families to active commute at least part of the way, promoting the use of park and walk/ride/scoot zones away from school grounds to reduce traffic. 2. Creating and promoting safe routes to schools and engage schools to promote their use. 3. Creating greater awareness of actual distances between home and school and the travel time for active modes. 4. Highlighting the benefit of students travelling to school carrying their school bags as an opportunity to be active while carrying a load, which could contribute to improve their muscular fitness [129].	1. Nationally representative data for primary and secondary students. 2. Data on the use of active transportation to other destinations. 3. Data on the use of multi-modal transport combining active transport with public transport. 4. Research about how far families and children are willing to travel using active transportation [129].
			In the Northern Territory, the Nightcliff Walk and Wheel initiative is aimed at encouraging students to walk and cycle to school. This is a local project in two dense suburbs lead by principals and parents from four schools. The project has a focus on roads safety for children and has implemented activities such as the Ride2School days, increasing cycling to school [129].		
D+	Wales	1	The report card mentioned the following initiatives led by charities to promote active travel to school: 1. Active Journeys—Sustrans School active travel program promotes active transportation through different actions like: (a) providing support to schools to develop active travel plans, (b) delivering activities and lessons, (c) offering free incentives to promote active travel, (d) providing resources and online travel challenges for the school community, and (e) rewarding schools with the School Mark award for achieving excellence in active and sustainable travel [130]. 2. Living Streets Walking initiatives: This charity has two main strategies for schools, the WOW Year-Round Walk to School challenge and the Five-Days Walking challenge. Both of these aim to engage primary and secondary students to walk to school encouraging them with an interactive travel tracker and the provision of incentives at the end of the challenge. This charity also encourages the celebration of the Walk to School Week in May every year [131].	Not reported	More research is needed on how children and young people travel to other places including shops, parks and friends’ or relatives’ houses [132].
D	Guernsey Channel Islands		Guernsey has an integrated transport strategy in place that promotes active travel with the aim of having a positive impact on the environment and the population’s health [133]. The On-island Integrated Transport Strategy aims to encourage active travel, followed by the use of public transport and to reduce the use of private motor vehicles. This strategy was initially planned to progressively advance to a taxation policy for high emission vehicles to support the promotion of active travel.	Through the implementation of the integrated transport initiative that supports active travel [133].	Not reported
			However, there are other actions in this strategy aimed at increasing active travel to school, such as: (a) Bikeability training at primary schools, (b) increasing the investments in walking and cycling infrastructure to improve safety for active commuters, (c) revising the speed limits to enhance the safety of vulnerable populations using active travel, and (d) developing and implementing travel plans for schools [134].		
D	Lebanon	1	Not reported	Not reported	Nationally representative samples for children and adolescents and on all active transportation means are required to gain a better understanding of this indicator [135,136].
D−	Canada	1	In Ontario, the Minister of Education expanded the funding for initiatives that improve the cognitive, physical, social and emotional well-being of students. Specifically walking school buses and biking-to-school programs have benefited from this increase in the funding [137]. These initiatives are part of the Ontario Active School Travel program (formerly Active & Safe Routes to School) which comprise five components: (a) education activities to foster the skills, confidence, and awareness such as workshops and route mapping. (b) Encouragement activities to inspire students, parents and school staff to try active travel modes. For example, walk and wheel seasonal events or walking school buses. (c) Engineering actions to create safe and accessible school sites, neighbourhoods and routes to school, such as school siting, signaling, parking restrictions, crosswalk improvements or crossing guards. (d) Enforcement of traffic policies to improve safety around schools. (e) Evaluations to measure the measure success, and demonstrate impact of the actions [138].	1. Creating a culture of active transportation, to make active transportation the norm. 2. Establishing lower speed limits in school areas. 3. Implementing traffic-calming devices (e.g., speed bumps/humps, chicanes, narrower intersections) to enhance compliance with speed limits, mainly in low-income areas, where more children engage in active transportation. 4. Hiring more crossing guards at busy intersections near schools. 5. Considering more progressive policies for low-income areas, to access to funding for active transportation interventions. 6. Considering children’s active transportation when planning schools and recreation facilities.	1. Research on activetransportation to destinations such as parks, stores, recreation facilities and other places. 2. Studies on children’s preferences for cycling and how to harness them in interventions. 3. More research is needed on how to facilitate children’s independent mobility. 4. More research is needed on the use of mixed modes of transportation to and from destinations [137].
			In 2017, three organizations in Canada (Canada Bikes, Green Communities Canada and the National Active and Safe Routes to School Working Group) created an active transportation alliance to advocate for the adoption and funding of a national active transportation strategy [137]. However, this strategy is not yet in place.	7. Encouraging schools to implement “drop-off spots” from which driven children could safely walk to school in groups [137].	
D−	United States	1	Safe Routes to School is a movement with initiatives at the regional, state and local levels that aims to promote walking and bicycling to school, improving safety, health and physical activity levels. Actions at the local level incorporate the six E’s integrated approach: (a) education through the provision of training in skills and knowledge to walk and bicycle safely and teaching the benefits of active transportation. (b) Encouragement to motivate children to travel actively through events and activities. (c) Engineering to improve streets and neighborhoods in order to make them more convenient for walking and bicycling. (d) Enforcement of safety regulation. (e) Evaluation of the success and opportunities to improve the initiatives in place. And (f) equity to ensure that the program benefits all demographic groups. Actions at the regional and state level are focused on finding funding and ensuring the proper use of the resources invested in the program. At the federal level, the Safe Routes Partnership advocates for policy and funding support and provides expert help, ideas and resources for the leaders at all levels [139].	1. Investing in infrastructure, programs, and policies that promote active transportation to and from school. 2. Allocating funding to provide and improve infrastructure to encourage active transportation (e.g., sidewalks, crosswalks, bike lanes, trails, etc.). 3. Encouraging children at home to use active transportation to school and other neighborhood locations. 4. Investing in infrastructure and policies such as Safe Routes to School and walking school buses. 5. Informing parents (at all income levels) about the benefits of active transportation [140].	1. The study of locations where children and adolescents walk and bike, duration and distance of trips as well as the main reasons for not engaging in active transportation. 2. Surveillance systems should include children under 12 years old [140].

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
