# Peer review of "Profiles of Active Transportation among Children and Adolescents in the Global Matrix 3.0 Initiative: A 49-Country Comparison"

_ijerph, 2020, doi:10.3390/ijerph17165997_

Round 1
Reviewer 1 Report
Thank you for letting me take part of this article concerning a very important aspect within health promoting research. The article is very well written and has a high significance and general interest.
I have really tried my best to come up with some suggestions for further improvement and the only things that I can think of is:
It is really impressive with 49 countries, however, there is more than 100 other independent states in the world and it would be interesting to hear your reflections about that.
It might be fruitful to relate your findings to the findings in "A future for the world’s children? A WHO–UNICEF–Lancet Commission", especially concerning their reasoning on Child flourishing index and Excess CO2 emissions.
Author Response
We appreciate your careful revision and suggestions to improve our manuscript. Please find attached the response to each of your reviews.

Reviewer 2 Report
In this paper the authors provide a thorough analysis of the Matrix 3.0 data in relation to active transport. The introduction is well written, easy to follow and provide evidence of why this is an important area of research and a knowledge gap which this article give input to.
The methods are thoroughly described and give a total overview of what is done in this study as well as previously in the Matrix project. The choices made by the authors are easy to follow and seems like the right choices to me.
The result is well described. The tables are generally good, however I have a small remark regarding table 4. The text referring to this table, line 6-10 could be presented in the same way for all three profiles to facilitate reading and comparing.
I also have some trouble following the figure and mean (based on scaled values) grading from -2 to 1. I think this could be clarified better.
The discussion is really good. I especially like that the limitations of the study is thoroughly discussed. The authors further discusses the need of future research in a good way.
I really want to congratulate the authors to this article, which I think will be frequently cited in articles regarding active transporting ahead.
Author Response

(The authors gave the same response as above.)

Reviewer 3 Report
Title: Active transportation among children and youth: a 49-country comparison
Dear authors,
The manuscript shows a novel and interesting approach informing about the active transportation patterns in young people along 49 countries, using a very accurate and already known methodology and providing to the scientific literature and to the society unique data and useful insights to improve the physical activity in young people through promoting active transportation behaviours.
The manuscript is outstanding. The strengths in the current manuscript are: a) it is the highest representation of countries under the paradigm of children and adolescents active commuting patterns, b) the writing is very clear and well organized that make an easier understanding, c) the implemented methodology is accurate and it has achieved to standardize data to have a common proposal regarding active commuting data in spite of the limitations, d) it provides very practical and useful information for future policies and it is derived directly from the obtained results, and e) it analyses 3 profiles is an accurate analysis to make more understandable the future policy proposals.
However, there are some major and minor issues to be addressed that may improve the manuscript:
- The distance home-school is the 1st predictor to active commute to school. The review studies state clearly the necessity of including this factor in the analysis and it can be a variable to provide information about the environment (i.e, compact city), policy (i.e., it is compulsory to go to the closest school) and even socioeconomic level (i.e, high families may drive their kids to a private school far away from home). So, I believe this variable may improve consistently the results and it may be included as a new latent variable in table 4, together with ATG, HDI and Gini Index. There are previous studies that have identified the threshold distance to walk to school in different countries, and that data could be used. In addition, some databases included in this analysis provide any type of distance or time home-school information. Including this variable, if possible, into the analysis could strength the profiles identified.
- The heterogeneity of the measures is a main limitation, but it is already explained there. In addition, I would include that in spite of the data belongs to each country, it is not representative of each country and it is suggested including information in method about how the data for each country was selected (because under my knowledge, there are other relevant databases for specific countries that have not been included there).
- This study provides new future implications for researching that should be clearly stated since they are relevant: a) to provide a common measurement (it has already been written), b) to include the distance home-school data and consider it in the statistical analysis, c) to provide a future research of how is the relation between active commuting to school and active commuting overall to have data about how is the contribution of commuting to school to the overall (i.e., %). d) to provide future quantitative measures (i.e., intensity, duration) of active commuting and not only the percentage of mode of commuting.
- Regarding the policies and practices identified by the expert groups of each country (table 5), some information should be provided for a better understanding. How was it selected? Was there a qualitative analysis to identify the policies? How was it categorized or grouped? It sounds that was opinion of experts, or did it come from previous validated works? It seems difficult to understand the validity of this information and how the diverse information for each country was joined into a same approach, regarding the environmental and cultural factors are diverse within each country. However, the information provided is of high interest since it helps to increase active commuting in the future.
Author Response

(The authors gave the same response as above.)

Reviewer 4 Report
General comment
I appreciate the opportunity to review this interesting paper. In this study the authors identified three different country profiles for active commuting from 49 countries. This paper is interesting and novel, and in my opinion, it would be of interest to readers of the International Journal of Environmental Research and Public Health. The main strengths of the manuscript are the novelty of the topic, which will allow to perform adapted active commuting strategies depending of the profile of the country, the inclusion of a big number of countries from the different continents and the accurate methodology to treat the data. However, there are some major and minor comments that have to be seriously taken into account before the manuscript is ready for publication. Below are my comments for the authors.
Major compulsory revision
- In the introduction section, the manuscript is focused about transportation, but in the last paragraph, the manuscript starts to be focused on transportation to school. Additionally, the included report cards are mainly focused in transportation to school (87%). I suggest to focus in the commuting to school behavior all the manuscript or clarify when you refers to commuting to different destination and commuting to school.
- Regarding the details related to active transportation in the report cards, it could of interest to include if, in the 49 countries, the National Education Act contains the active transportation to school as an educational content. This might be really important due to the work is mainly focused in active transportation to school in children and acolescents, and the main policy implementation for this population could be the National Education Act, allowing a deeper explanation of the obtained profiles. If this is not possible, this fact should be included as limitation and discussed.
- Regarding the table 3 information and due to almost all the countries report transportation to/from school, in the same way in which you stated that the number of report cards that provide details about sex, age group (i.e., children and adolescents) or commuting frequency, it is important to analyze and to discus the direction of the transportation to and/or from school. Previous studies have reported different modes of commuting to go to school compared with the coming back from school, and the transportation direction included in the report could influence to the data.
Minor Essential Revisions
- I suggest to use the expression “children and adolescents” along the manuscript instead of “children and youth”.
- In the abstract, in line 19, more information about the data analysis is necessary.
- In the abstract, in lines 19-20, some context about what means the “grade C” it is necessary.
- In keywords, some of them are repeated from title and/or abstract. I suggest using synonyms to increase the visibility of the work.
- In the method section, in the categorization of the Human Development Index, why did you include low and medium in the same category? Did you perform a sensitive analysis with low and medium HDI in different categories?
- In the table 3, in the column names, I suggest to use “Gender” instead of “Data gender”, and “Age” instead “by Age groups”.
- In table 4, include the number of countries with the percentage.
Author Response

(The authors gave the same response as above.)

Round 2
Reviewer 4 Report
Thank you for your answers. I do not have more comments to improve the manuscript.